# Primary Thrombophilia XVII: A Narrative Review of Sticky Platelet Syndrome in México

**DOI:** 10.3390/jcm11144100

**Published:** 2022-07-15

**Authors:** Claudia Minutti-Zanella, Laura Villarreal-Martínez, Guillermo J. Ruiz-Argüelles

**Affiliations:** 1Laboratorios RUIZ-Escuela de Ciencias Médicas, Universidad Popular Autónoma del Estado de Puebla (UPAEP), Puebla 72530, Mexico; claudia.minutti@upaep.edu.mx; 2Facultad de Medicina, Hospital Universitario de Nuevo León, Monterrey 64460, Mexico; drlaura.villarrealmtz@gmail.com; 3Centro de Hematología y Medicina Interna, Clínica RUIZ, Puebla 72530, Mexico

**Keywords:** thrombophilia, sticky platelet syndrome, hyperaggregability

## Abstract

Sticky Platelet Syndrome (SPS) is a disorder characterized by platelet hyperaggregability, diagnosed by studying in vitro platelet aggregation with ADP and epinephrine. It is the second most common cause of thrombophilia in Mexican Mestizos and manifests as an autosomal dominant trait which, combined with other coagulopathies, contributes significantly to the morbidity and mortality of patients with primary thrombophilia. It is easily treatable with antiplatelet drugs; however, the methods for diagnosis are not readily available in all clinical laboratories and the disorder is often overlooked by most clinicians. Herein, we present the results of more than 20 years of Mexican experience with the study of SPS in a Mestizo population.

## 1. Introduction

In 1995, Mammen and colleagues studied a hereditary condition of hyper adhesive platelets that clump upon standard surface contact, after studying a family whose members suffered from rare arterial thrombotic events without identifiable risk factors such as diabetes or atherosclerosis to predispose them and otherwise normal coagulation laboratory parameters. Later, they observed the same condition in more than 200 patients who, as named by Holiday and colleagues, suffer from “Sticky Platelet Syndrome (SPS)”. The condition is associated with angina pectoris, acute myocardial infarction, cerebral ischemic attacks or strokes, ischemic optic neuropathy, and recurrent venous thromboembolism, even while on optimal anticoagulant therapy [1].

We know now that SPS is a qualitative platelet disorder with familial occurrence that appears to have an autosomal dominant component as well, although not all patients have relatives with the disorder. It is characterized by increased platelet aggregation in response to ADP and epinephrine in vitro, and it is one of the most common causes of arterial thrombosis and pregnancy complications. Patients with SPS usually have their first thrombotic episode before age 40 and may or not have other acquired risk factors for thrombophilia. Anticoagulants such as vitamin K antagonists are inefficient, but some antiplatelet drugs have been shown to be effective at preventing rethrombosis and diminishing aggregation. An excellent review of the history of the disease and future perspectives was published by Kubisz and colleagues [2].

It has been considered that the glycoprotein receptors on the platelet surface may be involved in the pathogenesis of SPS; however, a specific cause of the condition has not been found [3]. Nonetheless, significant mortality and morbidity occur from these thrombotic events, such as paralysis, cardiac disability from repeated coronary events, miscarriages, and loss of vision and mobility. Rodger Bick (1998) mentioned that early diagnosis and treatment can prevent arterial and venous thrombotic events; however, clinicians and laboratories were unaware of the prevalence of the syndrome and thus failed to direct their diagnosis towards it. After studying 78 patients with said characteristics over a two year-period, he established that SPS is a common cause for arterial and venous thrombotic events in otherwise healthy patients [4]. These studies set the precedent for us to pioneer the study of the Mexican Mestizo population to determine the prevalence, incidence, and genetic background of SPS, aiming to determine its impact on morbidity and mortality in our country compared to other ethnic cohorts.

### 1.1. Primary Thrombophilia and Sticky Platlet Syndrome: The Mexican Experience

Pons-Estel and colleagues defined “mestizo” as those individuals born in Latin America who had both Amerindian and white ancestors, opposed to whites who have all white European ancestors and Amerindians, who have full autochthonous ancestry [5]. In 2005, Silva-Zolezzi and colleagues published the results of the first Mexican Genome Diversity Project (MGDP), which aimed to assess genetic ancestry in Mexicans to develop genomic medicine and genetic analysis in our country and Latin America. The study showed the great genetic diversity of Mexicans given by “mestizaje”, which aside from adding to the cultural richness and beauty of our country, poses a clinical challenge on the study of diseases and raises a question on whether results obtained from studies on Caucasian or other populations are truly applicable to ours [6].

Our first study related to the investigation of the primary causes of thrombophilia in Mexican Mestizos was carried out on 102 persons with clinical features of inherited thrombophilia, who were tested for the activated protein C resistance (APCr) genotype and phenotype as well as levels of coagulation proteins C and S, antithrombin, plasminogen, tissue type plasminogen activator activity, plasminogen activator inhibitor activity, plasminogen activator inhibitor type I, anti-phospholipid antibodies and lupus anticoagulants. While 46% of the patients fit within the normal range for all tests, 39.2% were consistent with the APCr phenotype and only 4% with the factor V Leiden mutation. This finding is relevant because studies on Caucasian groups report that 20–60% of cases present such mutation, whereas in Mexican Indian groups it is almost nonexistent. In this cohort, most of the cases were acquired or unrelated to the factor V mutation, which led to conclude that the ethnic composition of Mexican Mestizo ancestry plays an important role in the etiology of primary thrombophilia [7].

In a continuing study, 37 Mexican Mestizo patients and 50 normal controls were tested under the same criteria to investigate prevalence of known mutations associated with primary thrombophilia. Four were heterozygous for the factor V Leiden mutation, 16 for the MTHFR 677 mutation, and 5 for the prothrombin 20,210 mutation. 6 were homozygous for the MTHFR 677 defect. It was also found that four individuals were compound heterozygotes for combinations of these mutations. MTHFR mutation alone is not sufficient to cause thrombophilia unless it is associated with other thrombophilia-causing conditions. Again, this study showed that the prevalence of mutations in Mexican Mestizos differs from that reported in Caucasians and paved the way for further analysis of these genetic differences and their implications on the diagnosis and prognosis of primary thrombophilia in Mexico [8].

In 2002, 10 patients with clinical characteristics of primary thrombophilia from the same ethnic group were prospectively studied to assess the prevalence of SPS. Platelet aggregation was measured from peripheral blood samples with increasing concentrations of ADP and epinephrine, while other coagulation and hemostasis parameters were measured as in previous studies [7,8]. Six out of ten patients fit within the SPS abnormality: five of them displayed other thrombophilia conditions linked to the genetic mutations previously studied, whereas only one presented SPS as a sole condition. Four of these six patients had a family history of thrombophilia. The study showed that SPS is frequently found in Mexican Mestizos with clinical characteristics of thrombophilia along with other genetic conditions that contribute to a multifactorial disease, changing the way clinicians approach diagnosis and treatment due to the implications on the health of patients and their families [9].

Given that few Mexican Mestizo patients with APCr-linked thrombophilia were affected by the FV Leiden mutation, the prevalence of other mutations such as HR2 haplotype, FV Cambridge, Hong Kong, and Liverpool was also looked at. Thirty-nine patients, regardless of their APCr phenotype status, were accrued for the study; inclusion criteria considered early-age and recurrent thrombosis, thrombosis at unusual anatomic sites, resistance to conventional therapy, and at least one episode of venous thrombosis, confirmed by phlebography or Doppler. Overall, 10% of patients were heterozygous for the FV Leiden mutation, 28% displayed the HR2 haplotype, one patient presented the Hong Kong mutation, and none presented the Cambridge or Liverpool mutations. This study yet again evidenced the differences in the genetic background of thrombophilia patients across ethnicities and concluded that the studied polymorphisms are nor relevantly implicated in thrombophilia in Mexican Mestizos [10].

In 2005, 46 Mexican Mestizo patients were accrued to assess the prevalence of SPS, protein C resistance, protein C activity and antigen, protein S, antithrombin, plasminogen, tissue type plasminogen activator activity, IgG and IgM isotypes of antiphospholipid antibodies, Factor V gene mutations, MTHFR 677 mutation and G20210A polymorphism. A total of 8% of individuals did not display any abnormality and thus were not counted for the final evaluation; 12% showed one abnormality and 88% percent presented two to five co-existing abnormalities; 48% of patients had SPS, 24% aPCR phenotype, 11% the FV Leiden mutation, 24% antiphospholipid antibodies, 9% protein S deficiency, and 13% protein C deficiency; 24% patients presented the HR2 haplotype and only one patient presented the Hong Kong mutation, supporting the results of the previous study [11]. Abnormalities in APCr were not significantly associated with the SPS phenotype in another study [12].

### 1.2. Multifactorial Thrombophilia

Hypercoagulability is a major health problem and has a high mortality and morbidity around the world. Inherited hypercoagulable states are associated with venous thrombosis rather than arterial problems, which are mostly due to the increased activation of platelets in the endothelial surface. Although genetic predisposition is unlikely to be the sole cause of a thrombotic event, people who have inherited more than one thrombophilia are at greater risk of thrombosis than those who are affected only by a single factor. Other “triggers” are needed to develop a thrombotic event, for example, atherosclerosis, pregnancy, or the use of oral contraceptives. This poses a model of “thrombosis threshold”, which is based on a combination of inherited hypercoagulable states and patient lifestyle [13]. Based on this, in 2007 we investigated the relationship between clinical markers and thrombophilia in 100 patients. Overall, 19% of patients presented only one abnormality, while 81% presented two or more. SPS was found in 57 patients and the presence of other mutations was consistent with previous studies. Results also showed an association between the FV Leiden mutation and resistance to activated protein C; meaning that 94% of Mexican Mestizos that had at least one clinical marker of thrombophilia develop a thrombophilic condition. However, it is yet more common that the development of these pathologies is enhanced by two or more coexisting thrombophilia conditions [14].

Moreover, thrombophilia is often concomitant in myeloproliferative disorders (MPDs) such as polycythemia vera, essential thrombocythemia, idiopathic myelofibrosis, and chronic leukemias. Since myeloproliferative diseases arise from acquired genetic mutations, the group studied 77 Mexican Mestizos with primary thrombophilia to look for the JAK2 V617F mutation (commonly associated with MPDs) aiming to find whether underlying MPDs could be the precipitating factor of thrombotic episodes. None of the patients carried the mutation and only four were found to have splanchnic thrombosis, concluding that MPDs are an improbable cause of thrombophilia in this population. It is also noteworthy that the prevalence of MPDs in Mexican Mestizos is lower than in Caucasians [15].

Later on, we studied the effect of SPS in pregnancy. A total of 268 Mexican Mestizo patients were studied, of which 108 female patients were selected for further analysis; 71% of these patients had been pregnant at some point, and 37% had experienced at least one spontaneous abortion. Within the subset of patients who suffered from a spontaneous abortion, 86% had SPS, while the remaining percentage were heterozygous the MTHFR mutation. At the time, data in Mexico indicated that 12–13% of pregnancies in the general population end in a spontaneous abortion, meaning that the relative risk of having a miscarriage is 2.66 times higher in women with SPS. Timely and efficient treatment with antiplatelet drugs in these patients could reduce the risk of obstetric complications significantly. This study evidenced the necessity of further investigating complications and treatment of SPS in pregnant women [16]. Sokol and colleagues mentioned that SPS is especially relevant in the clinical management of patients with recurrent abortion [17].

### 1.3. Subtypes of SPS and Inheritance

It has been proposed that SPS is an autosomal dominant inherited disease that can manifest in one of three ways: type I, characterized by platelet hyperaggregability with ADP and epinephrine; type II, where aggregation happens only with epinephrine; and type III, only with ADP. To further study this observation, the group studied 5 kindreds of patients with known SPS and previous thrombotic episodes. A complete laboratory workup for thrombophilia was performed and in all five kindreds, where other relatives presented SPS. Results showed that family members of patients who carried the MTHFR mutation also had the mutation, and in one kindred, it was found in members of different generations [18]. Further on, the group aimed to establish a correlation between SPS phenotype and the GPIIIa PL A1/A2 polymorphism, a known marker of hyperaggregability and thromboembolism. A total of 160 patients with a clinical marker of primary thrombophilia were studied, of which 95 presented SPS (61 patients with type 1, 6 and 28 with type 2 and type 3, respectively). Of these patients, 79 had the PL A1/A2 genotype, 15 displayed the A1/A2 genotype, and 1 showed the PL A1/A2 genotype. In healthy controls, the frequencies were similar; thus, it was inferred that there is no significant association between PL A1/A2 polymorphisms and SPS phenotype [19].

In another study performed on 86 patients, we found that 65% of the cases were SPS type I, 10% type II, and 25% type III. Venous thrombosis was more frequent than arterial with 70% of cases, presenting in the lower limbs, CNS, upper limbs, mesenterial veins, and retina. There was no association between SPS type and localization of the thrombi, and no correlation between gender and localization of the episode nor subtype of SPS. Once again, this showed that there are important epidemiological differences between SPS in Mexican Mestizos and Caucasians. Caucasians suffer from SPS type II more frequently [20], whereas type I is more frequently seen in Mexican Mestizos. SPS is also the second most common cause of thrombophilia in this ethnic group [21]. We also conducted a study to define the identification of SPS worldwide and found that patients with the condition have been identified and reported in the five continents [12].

### 1.4. Insights on the Treatment of SPS

Treatment of SPS consists of diminishing platelet hyperaggregability with antiplatelet drugs such as aspirin. To evaluate the efficacy of this treatment on the prevention of rethrombosis and hyperaggregability, 55 patients with at least two assessments of SPS phenotype were treated and followed for up to 129 months with platelet aggregation studies. A total of 40 patients were treated with aspirin, 13 with a combination of aspirin and clopidogrel, and 2 with clopidogrel only. Two of these patients developed another vaso-occlusive episode in the retinal central artery despite treatment after 52 and 129 months; however, they showed no additional thrombophilia-causing conditions besides SPS after a full laboratory workup (Velázquez-Sanchez-de-Cima et al., 2013). Since SPS can contribute to “multifactorial thrombophilia”, it is important that key laboratory tests are performed. Nevertheless, it was observed that treatment with common antiplatelet drugs such as aspirin and/or clopidogrel was effective in 96.4% of patients with SPS, regardless of the subtype. The results are in concordance with those obtained in a larger cohort (*n* = 270) by Kubisz and colleagues [22]. The findings of this study and observations made in the Mexican Mestizo population were presented at the 18th International Meeting of the Danubian League against Thrombosis and Haemorrhagic Disorders in 2015 and this continues to be the treatment of choice nowadays [23].

To further analyze the features of the treatment of SPS worldwide, we conducted a meta-analysis of 108 papers containing the term “Sticky Platelet” in the title or the abstract obtained from PubMed; of these, 43 were selected and 1783 patients with the condition were identified. We found that 332 patients received antiplatelet drugs, of which 303 were given aspirin only, 29 received combinations of heparin or coumadin with aspirin, and 2 patients received heparin + alteplase or abciximab. The rate of rethrombosis on these patients was 1.5%, showing that physicians around the world are aware that the use of antiplatelet drugs in SPS patients is beneficial. Although the treatment has been proven to be effective to control the condition, the importance of investigating its pathophysiology and epidemiology around the world has been deemed extremely important [24].

### 1.5. Concluding Remarks

Since the first description of Sticky Platelet Syndrome, it has been within our uttermost interest to contribute to the study of the genetic factors, clinical management, and diagnosis of the disease; especially given that it seems to be the second most common cause of thrombophilia in our country. SPS does not usually lead to thrombosis on its own, but rather needs association with another thrombophilia-causing condition to manifest itself, such as estrogen use, mutations, other alterations in blood coagulation, and in most recent times, COVID-19 [25,26]. In some patients, SPS can be so insidious that clinicians often mislabel thrombotic events as idiopathic. However, in patients with a high level of baseline genetic hypercoagulability, simple triggers could initiate thrombotic episodes and make them more likely to recur [11].

Even though the exact pathophysiology of SPS is not fully understood to date, a detailed review of known platelet function and molecular and genetic features associated with this syndrome was recently published by García-Villaseñor and colleagues. Some studies have been performed to assess the implications of microRNAs in platelet function, as well as to establish the relationship of polymorphisms with SPS complications, most of which are related to chronic degenerative diseases and infertility [27].

There is an urgent need to identify the cause of SPS, given that it is the most frequent cause of hereditary thrombophilia in México and probably in other countries as well. The gold standard to diagnose SPS is aggregometry; but since this technique is not readily available in most clinical laboratories, it is important to study more specific markers that could be studied with ease [28]. Many clinical professionals believe that SPS is only a reflection of “laboratory artifacts”; however, this is not due to lack of proof, but rather lack of understanding and availability of diagnostic methods. Given that SPS testing needs recently collected blood specimens, it is not possible to refer samples for testing, highlighting the need to standardize laboratory protocols and provide trustworthy information to make it more readily available for patients and clinicians. In Mexico, the topic is worthy of study since SPS has been proven to go hand-in-hand with other causes of thrombophilia, the most frequent cause of spontaneous miscarriages in thrombophilic women, and an important cause of venous thromboembolism. Nonetheless, SPS is an easily treatable condition given that antiplatelet drugs are cheap, available, and effective. Therefore, knowledge of the condition implies improving the quality of life for patients with this and other concomitant disorders; see Box 1 and Figure 1.

Box 1Salient features of Sticky Platelet Syndrome.(1)SPS is a phenotype of platelet hyperaggregability,
defined by increased in vitro platelet aggregation after the addition of very
low concentrations of adenosine diphosphate and/or epinephrine. The
concentrations and dilutions of the agents are relatively well standardized.(2)The genotype is currently unknown, but several
observations on the genes of platelets proteins are being studied: platelet
glycoprotein IIIa PLA1/A2; platelet glycoprotein 6, growth arrest specific 6,
coagulation factor V, integrin subunit beta 3, platelet endothelial
aggregation receptor 1, serpin family C member 1, serpin family E member 1.(3)The SPS phenotype is probably the expression of genetic
conditions interacting with other medical conditions or environmental
factors, such as diabetes mellitus, hormonal therapy, and pregnancy.(4)SPS may lead into both arterial and venous thrombosis,
the latter being more frequent.(5)SPS is a hereditary autosomal dominant trait.(6)SPS is the most frequent cause of hereditary
thrombophilia in México, and probably in other countries.(7)Patients with SPS have been identified and treated in all continents of the world.(8)SPS is a frequent cause of miscarriages and obstetric complications.(9)SPS usually needs another thrombophilic condition to
fully express as a thrombotic episode. It has recently been described as a
risk factor for thrombosis during COVID-19.(10)The hyperaggregability of SPS reverts to employing
antiplatelet drugs and the re-thrombosis rate of persons with the syndrome is
very low while being on treatment. Most patients revert the
hyperaggregability with aspirin, but around one quarter need two antiplatelet
drugs. It is therefore advisable to assess the SPS phenotype after starting
the antiplatelet drug, in order to define further treatment. Treating persons
with SPS with oral anticoagulants does not reduce the re-thrombosis rate(11)Claiming that SPS is a non-entity indicates that it is
not being assessed properly and may also be detrimental for patients. The
treatment is cheap, available and effective, as well as tolerated by most
persons, which is the use of low-doses of aspirin and other antiplatelet drugs.

## Figures and Tables

**Figure 1 jcm-11-04100-f001:**
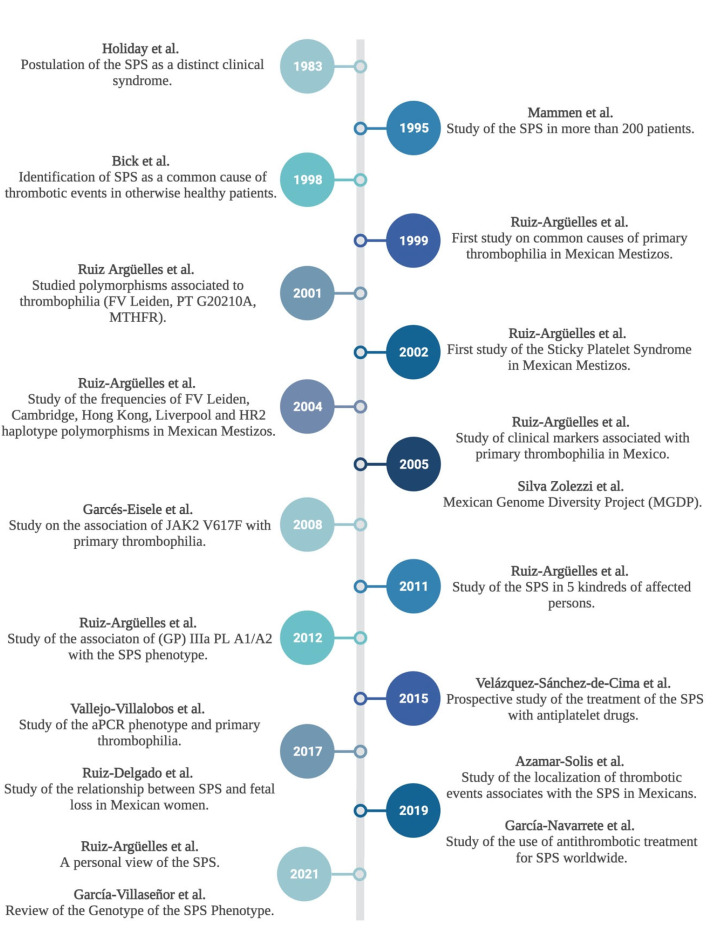
Timeline of the initial studies on Sticky Platelet Syndrome, and subsequent studies conducted and published in México. SPS: Sticky Platelet Syndrome. MTHFR: Methylenetetrahydrofolatereductase. (GP)IIIa PL A1/A2: Glycoprotein IIIa. aPCR: activated protein C resistance. Created in Biorender.com (accessed on 2 July 2022).

## Data Availability

Not applicable.

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
