# Peer review of "Primary Thrombophilia XVII: A Narrative Review of Sticky Platelet Syndrome in México"

_jcm, 2022, doi:10.3390/jcm11144100_

Round 1

Reviewer 1 Report

There are a number of minor issues throughout as follows:

1. Page 1: lines 28-30: "The condition associated with angina pectoris, acute myocardial infarction, cerebral ischemic attacks or strokes, ischemic optic neuropathy, and recurrent venous thromboembolism, even while on optimal anticoagulant therapy." The condition is associated...

2. Page 2, lines 67-68: "who were tested for the activated protein C resistance (APCr) genotype as well as levels of coagulation proteins C and S, AT-III,... The authors later state (lines 71,72): "...39.2% were consistent with the APCr phenotype and only 4% with the factor V Leiden mutation." So, authors must have studied both the APCr phenotype and genotype? Second issue that AT-III undefined, and these days just called 'antithrombin'.

3. Page 3; lines 114-123. The authors here have used the full term 'antithrombin III' (line 114) which should probably just be 'antithrombin'. Also, authors write protein C and protein S (line 114) and then use PC and PS (line 120) without pre-defining as protein C and protein S; probably better just to use protein C and protein S throughout, as PS and PC are not used later. Finally, authors write 'sticky platelet syndrome' (line 123) but have already defined as SPS. Best to use the defined abbreviation once defined.

4. Line 136: "obtaining similar results" needs clarification; similar to what?

5. Line 139: "between the Leiden mutation" clarify as FV Leiden, since there are other 'Leiden mutations' described.

6. Lines 140/141: "clinical marker of thrombophilia present thrombophilia." does not make sense; revise for clarity.

7. line 142: Word 'propitiated' - not sure this is an appropriate context for usage.

8. line 148: "aiming find out" - aiming to find out

9. Lines 172/173: "Results showed that family members of patients that carried the MTHFR mutation also presented it, and in one kindred it was found in members of different generations." meaning of 'it' unclear - presumably SPS?

10. Lines 182/183: "SPS type I, 10% type II and 25% type III." Previously identified groups as type 1, 2 and 3. Authors need to be consistent

11. lines 188-189: "Caucasians suffer from SPS type II more frequently, 188 whereas type I is more frequently seen in Mexican Mestizos." As above; use 1, 2 or I, II but not both. Also, statement of fact requires a reference.

12. Line 189/190: "SPS is also the second cause  of thrombophilia in this ethnic group" second most common cause?

13. Line 238/239: "the second cause of thrombophilia in Mexico" second most common cause?

14. Table 1: "The SPS is the most frequent cause of hereditary thrombophilia in in México and probably in other countries." Could also be mentioned in the main text, where authors state second most common cause of thrombophilia.

15. Table 1 line 226: "frequent cause on miscarriages" - frequent cause of miscarriages

16. Line 277: "detrimental por the patients" - "detrimental for the patients"

17. line 277: "since the consequences of defining is a simple, cheap and effective treatment" ?defining SPS?

18. figure 1 legend: "Timeline of the studies conducted and published in México" the first 3 entries were outside of Mexico, but reflect important historical context. Suggest rename legend as "Timeline of the initial studies on Sticky Platelet Syndrome, and subsequent studies conducted and published in México"

Author Response

Dear Reviewer 1,

We thank you so much for the revision of our manuscript and for the suggestions that were pointed out. They indeed helped us convey our message better and add to the quality of our paper. We believe it is not necessary to discuss each of the changes made point by point, since all the grammatical errors you pointed out were changed accordingly as suggested.

A reference was added to justify the fact that SPS type II is most frequently observed in Caucasians:

Skerenova M, Jedinakova Z, Simurda T, Skornova I, Stasko J, Kubisz P, et al. Progress in the Understanding of Sticky Platelet Syndrome. Seminars in Thrombosis and Hemostasis 2016. 43,008-013.

Also, the numerals for SPS types I, II and III were unified accordingly across the text.

Reviewer 2 Report

- authors should provide a table gathering the main findings from the literature. Please update.

- authors should also include a representative figure for this paper in order to improve the readability of the content of the manuscript. This will give a more comprehensive overview of the paper.

Author Response

Dear Reviewer 2, 

We appreciate your kind comments and thank you for your suggestions. We believe that table 1 describes the key features of SPS known to the moment and figure 1 includes all of the studies that we included in the text. Therefore, adding another table or figure with the findings could be redundant since the information is already displayed there and summarized in the text. Below we attach the corrected manuscript with the changes suggested by Reviewer 1. We hope that you find it well and we thank you in advance for your time and collaboration.